# Assays for the Evaluation of the Immune Response to Marburg and Ebola Sudan Vaccination—Filovirus Animal Nonclinical Group Anti-Marburg Virus Glycoprotein Immunoglobulin G Enzyme-Linked Immunosorbent Assay and a Pseudovirion Neutralization Assay

**DOI:** 10.3390/vaccines10081211

**Published:** 2022-07-29

**Authors:** Thomas L. Rudge, Nicholas J. Machesky, Karen A. Sankovich, Erin E. Lemmon, Christopher S. Badorrek, Rachel Overman, Nancy A. Niemuth, Michael S. Anderson

**Affiliations:** 1Battelle, West Jefferson, OH 43162, USA; macheskyn@battelle.org (N.J.M.); sankovichk@battelle.org (K.A.S.); lemmon@battelle.org (E.E.L.); niemuth@battelle.org (N.A.N.); andersonms@battelle.org (M.S.A.); 2Contract Support for the U.S. Department of Defense (DOD) Joint Program Executive Office for Chemical, Biological, Radiological, and Nuclear Defense (JPEO-CBRND) Joint Project Manager for Chemical, Biological, Radiological, and Nuclear Medical (JPM CBRN Medical), Fort Detrick, MD 21702, USA; christophe.s.badorrek.ctr@army.mil; 3U.S. Department of Defense (DOD) Joint Program Executive Office for Chemical, Biological, Radiological, and Nuclear Defense (JPEO-CBRND) Joint Project Manager for Chemical, Biological, Radiological, and Nuclear Medical (JPM CBRN Medical), Fort Detrick, MD 21702, USA; rachel.l.overman.civ@army.mil

**Keywords:** Marburg virus, Sudan virus, ELISA, pseudovirion neutralization assay

## Abstract

Since the discovery of the Marburg virus (MARV) in 1967 and Ebola virus (EBOV) in 1976, there have been over 40 reported outbreaks of filovirus disease with case fatality rates greater than 50%. This underscores the need for efficacious vaccines against these highly pathogenic filoviruses. Due to the sporadic and unpredictable nature of filovirus outbreaks, such a vaccine would likely need to be vetted through the U.S. Food and Drug Administration (FDA), following the Animal Rule or similar European Medicines Agency (EMA) regulatory pathway. Under the FDA Animal Rule, vaccine-induced immune responses correlating with survival of non-human primates (NHPs), or another well-characterized animal model, following lethal challenge, will need to be bridged for human immune response distributions in clinical trials. A correlate of protection has not yet been identified for the filovirus disease, but antibodies, specifically anti-glycoprotein (GP) antibodies, are believed to be critical in providing protection against the filovirus disease following vaccination and are thus a strong candidate for a correlate of protection. Thus, species-neutral methods capable of the detection and bridging of these antibody immune responses, such as methods to quantify anti-GP immunoglobulin G (IgG)-binding antibodies and neutralizing antibodies, are needed. Reported here is the development and qualification of two Filovirus Animal Nonclinical Group (FANG) anti-GP IgG Enzyme-Linked Immunosorbent Assays (ELISAs) to quantify anti-MARV and anti-Sudan virus (SUDV) IgG antibodies in human and NHP serum samples, as well as the development of pseudovirion neutralization assays (PsVNAs) to quantify MARV- and SUDV-neutralizing antibodies in human and NHP serum samples.

## 1. Introduction

Filoviruses are negative-sense, single-stranded RNA viruses belonging to the Filoviridae family [1,2]. This family currently consists of five genera: *Marburgvirus*, *Ebolavirus*, *Cuevavirus*, *Striavirus*, and *Thamnovirus*, with a sixth proposed as *Dianlovirus* [3]. Only viruses in the *Marburgvirus* and *Ebolavirus* genera have been documented to cause severe disease in humans and nonhuman primates (NHPs), the Marburg disease and Ebola disease, respectively [4]. Filoviruses are zoonotic in nature, causing primary infections in humans visiting mines or caves inhabited by the African fruit bat (*Rousettus aegyptiacus*), the natural reservoir of Marburg virus (MARV), or exposed to infected NHPs. Secondarily, transmission from human-to-human can occur through direct contact of mucous membranes or skin lesions with infected blood, secretions, organs, or other bodily fluids or through contact with contaminated materials (e.g., clothing, bedding, needle sticks). For both the Marburg disease and Ebola disease, the symptoms and disease manifestation are similar. The incubation period can be from 2 to 21 days post-exposure with initial symptoms consisting of high fever, fatigue, muscle and joint pain, and a sore throat, followed by nausea, vomiting, abdominal pain, and diarrhea. As the disease progresses, decreased liver and kidney function and internal and external bleeding are observed. There may also be central nervous system involvement, resulting in confusion, irritability, and aggression. Death usually occurs between Days 8 and 9 after symptom onset due to dehydration and organ failure [5]. There is currently a licensed vaccine for Ebola virus (EBOV) (rVSV-ZEBOV-GP, Ervbro, Merck) [6] with several promising candidates in the pipeline [7,8]; however, there is currently no licensed vaccine or therapeutic for MARV or Sudan virus (SUDV), only supportive care. There are, however, several candidate vaccines for both MARV and SUDV in various stages of pre-clinical and clinical development [7,9,10,11]. Both MARV and SUDV have been classified as Select Agents by the Centers for Disease Control and Prevention (CDC), as Category A Priority Pathogens by the National Institute of Allergy and Infectious Diseases (NIAID), and as Risk Group 4 agents by the World Health Organization (WHO). Additionally, working with the live virus is restricted to Biosafety Level 4 (BSL-4) laboratories.

The genus Marburgvirus has a single species, *Marburg marburgvirus*, consisting of MARV and Ravn virus (RAVV). The first outbreak of Marburg disease occurred in 1967 simultaneously in Marburg and Frankfurt, Germany, and in Belgrade, Serbia, following exposure to African green monkeys (Cercopithecus aethiops) infected with MARV, originating from Uganda and shipped to all three locations [12,13]. The purpose of these animals was to provide kidney cells for the propagation of poliomyelitis vaccine strains. Since then, there have been 13 MARV outbreaks and sporadic cases, mainly in Africa, with 466 confirmed cases and 376 deaths, and a case fatality rate near 80%. Most of these cases occurred in Sudan in 2004–2005 with 252 confirmed cases and 227 deaths (90% case fatality rate) [14,15].

The genus Ebolavirus has six species: *Bombali ebolavirus* (BOMV), *Bundibugyo ebolavirus* (BDBV), *Reston ebolavirus* (RESTV), *Sudan ebolavirus* (SUDV), *Tai Forest ebolavirus* (TAFV), and *Zaire ebolavirus* (EBOV). Only BDBV, SUDV, TAFV, and EBOV are known to have caused outbreaks in humans, with primary infections caused by spillover from wildlife reservoirs followed by secondary infections caused by human-to-human transmission [16]. The first emergence of the Ebola disease was two almost simultaneous outbreaks caused by SUDV in Sudan and EBOV in Yambuku in the Democratic Republic of the Congo in 1976 [17,18]. Since then, there have been more than 35 Ebola disease outbreaks, causing more than 15,000 deaths, with a case fatality rate near 44%. Specifically for SUDV, there have been eight outbreaks, causing more than 750 deaths and a case fatality rate near 50% [14,19].

The Filovirus glycoprotein (GP) is expressed on the exterior of the viral particle, is required for virus binding and entry into the cytoplasm of susceptible host cells and is a primary target for neutralizing and protective antibodies [20,21,22,23,24,25,26]. During the 2014–2016 Ebola disease outbreak in Sierra Leone, higher levels of EBOV anti-GP-specific IgG at one week after the onset of symptoms were correlated with survival in 65 confirmed cases [27]. Therefore, an efficacious vaccine candidate for MARV and SUDV will likely require strong induction of GP-specific antibodies. The Ervbro vaccine and other candidate vaccines that have been moved into clinical trials present the EBOV GP as an antigen and induce anti-GP antibodies in NHPs [28,29,30,31,32,33] and humans [34,35,36,37,38,39,40,41,42,43,44,45,46,47,48,49,50]. Thus, anti-GP IgG antibodies (both binding and neutralizing) are the likely front-runners for a correlate of protection [51].

Due to the sporadic and unpredictable nature of filovirus outbreaks, the high mortality associated with infection, and the difficulty in performing efficacy studies in humans, the efficacy of MARV and SUDV vaccines will likely require evaluation under the FDA Animal Rule [52,53,54]. The Animal Rule provides a pathway to vaccine regulatory evaluation when it is unethical or not feasible to conduct human efficacy studies, such as in the absence of large natural outbreaks. To use the Animal Rule, immune responses that correlate with survival in well-characterized animal models must be bridged for the distribution of immune responses in placebo-controlled human vaccine clinical trials to establish human efficacy. Species-neutral immunological methods are ideal for bridging data between humans and animal models.

Here we describe the methods and development of two assays, an ELISA to quantify anti-GP IgG levels and a PsVNA to quantify neutralizing antibodies. The Filovirus Animal Nonclinical Group (FANG) [55] anti-MARV GP and anti-SUDV GP IgG ELISAs have been fully developed and qualified for quantitation of anti-MARV and anti-SUDV GP IgG antibodies in human and NHP serum. Both ELISAs are based on the previously-published FANG anti-EBOV GP IgG ELISA and follow the same method but use a different recombinant GP as the coating antigen (specific for MARV or SUDV) [56]. The resulting ELISAs reproducibly quantify levels of anti-MARV/SUDV GP IgG antibodies in human and NHP serum samples from MARV/SUDV disease survivors and vaccinated individuals (human and NHP). The PsVNA has been fully developed and is able to reproducibly quantify levels of neutralizing antibodies in humans and NHPs for both MARV and SUDV. Both assay readouts are strong candidates for a correlate of protection for the evaluation of candidate MARV and SUDV vaccines.

## 2. Materials and Methods

### 2.1. Anti-MARV and Anti-SUDV GP IgG ELISA Procedure

The validated anti-EBOV GP IgG ELISA method [56] was used as a template for the anti-MARV and anti-SUDV GP IgG ELISA method development. All buffers, incubation periods, temperatures, and handling procedures were the same as those used for the anti-EBOV GP IgG ELISA.

To calculate the anti-MARV or anti-SUDV GP IgG concentration for a sample, a 4-Parameter Logistic (4PL) curve was first fit to the OD values obtained for the RS. The concentrations (based on the acceptance criteria described in [56]) from each dilution of the six-point dilution series of a TS were then calculated from the RS using the 4PL equation. The reported concentration for each TS was the arithmetic average of the values from the acceptable dilution points.

An illustration of the general ELISA method is presented in Figure 1 and the ELISA plate layout is presented in Figure 2.

### 2.2. Recombinant Marburg Virus (MARV) Glycoprotein

Purified rGP with an amino acid sequence corresponding to the GP from MARV Ci67 (Accession Number ADM72998.1) was produced at Advanced Bioscience Laboratories, Incorporated (ABL, Rockville, MD, USA), Lot numbers 13DEC2013 and 26APR2016, with concentrations of 1.0 mg/mL and 1.3 mg/mL, respectively. The rGP is on a stability testing plan for total protein concentration and protein purity and has passed all stability criteria following storage at ≤−70 °C for at least 91 months (Lot 13DEC2013) and 36 months (Lot 26APR2016).

### 2.3. Recombinant Sudan Ebolavirus (SUDV) Glycoprotein

Purified rGP with amino acid sequence corresponding to the GP from SUDV Gulu (Accession Number AAU43887.1) was produced at ABL, Lot number 22JUL13, with a concentration of 1.08 mg/mL. The rGP is on a stability testing plan for total protein concentration and protein purity and has passed all stability criteria following storage at ≤−70 °C for at least 91 months.

### 2.4. Panels of Sera from Convalescent MARV and SUDV Infection Survivors

Panels of sera from the convalescent survivors of a previous MARV or SUDV infection were received from Dr. John Dye (United States Army Medical Research Institute of Infectious Diseases, USAMRIID, Fort Detrick, MD, USA) for use in the creation of the first-generation critical reagents (RS and QCs). These panel samples (MARV samples = MK1, MK2, MK3, MK4, MK5, MK6, MK8, and MK9) (SUDV samples = G12, G16, G21, G23, G24, G26, G28, G44, G54, L1, and L2) were evaluated to determine a pooling strategy for the first-generation RS and QCs. Results are summarized for the MARV and SUDV (Appendix A). The generation of the RSs and QCs using these samples is described in the following sections.

### 2.5. Human Reference Standard Serum

There have been three generations of MARV RSs and two generations of SUDV RSs qualified for use in the ELISAs (Appendix A).

#### 2.5.1. MARV Reference Standard

The first-generation MARV RS (Lot Number BMIMARV101) was a pool of sera from two convalescent MARV infection survivors received from the USAMRIID that was diluted with pooled naïve human serum. Starting dilutions of 1:50 and 1:100 and serial dilutions of 1:2, 1:1.9, 1:1.8, 1:1.7, 1:1.6, and 1:1.5 were evaluated, and the resulting OD values were used to generate RS curves from a 4PL model. The RS was assigned an arbitrary concentration of 1000 ELISA units/mL.

The second-generation MARV RS (Lot Number BMIMARV104) was also a pool of sera from two convalescent MARV infection survivors received from the USAMRIID that was diluted with pooled naïve human serum. BMIMARV104 was deemed to be parallel to BMIMARV101 and was assigned an anti-MARV GP IgG concentration of 935.95 ELISA units/mL following more than 100 independent evaluations using BMIMARV101 as the RS.

The third-generation MARV RS (Lot Number BMIMARV107) was created by diluting purified human IgG polyclonal anti-MARV antibodies derived from a transchromosomal (Tc) bovine vaccinated with MARV VLP (Musoke strain) antigens (IBT BioServices Cat# 0566-001, Rockville, MD, USA) (SAB-170M Anti-Marburg Antibody, Lot #RM1801292MV) with naïve human serum. BMIMARV107 was deemed to be parallel to BMIMARV104 and was assigned an anti-MARV GP IgG concentration of 1057.89 ELISA units/mL following more than 100 independent evaluations using BMIMARV104 as the RS.

#### 2.5.2. SUDV Reference Standard

The first-generation SUDV RS (Lot Number BMISUDV101) was a pool of sera from two convalescent SUDV infection survivors received from the USAMRIID. Starting dilutions of 1:50 and 1:100 and serial dilutions of 1:2, 1:1.9, 1:1.8, 1:1.7, 1:1.6, and 1:1.5 were evaluated, and the resulting OD values were used to generate RS curves from a 4PL model. The RS was assigned an arbitrary concentration of 1000 ELISA units/mL.

The second-generation SUDV RS (Lot Number BMISUDV104) was created by diluting purified human IgG polyclonal anti-SUDV antibodies derived from a Tc bovine vaccinated with SUDV VLP (Gulu strain) antigens (IBT BioServices Cat# 0567-001) (SAB-170S Anti-Sudan Antibody; Lot # RM1801341SV) with naïve human serum. BMISUDV104 was deemed to be parallel with BMISUDV101 and was assigned an anti-SUDV GP IgG concentration of 1143.22 ELISA units/mL following more than 100 independent evaluations using BMISUDV101 as the RS.

### 2.6. Human Quality Control Serum

There have been three QC generations each of serum with high anti-MARV GP IgG levels (QC-High) and with low anti-MARV GP IgG levels (QC-Low) generated during the development/qualification of the MARV ELISA (Appendix A). Likewise, there have been two QC generations each of QC-High and QC-Low generated during the development/qualification of the SUDV ELISA (Appendix A).

#### 2.6.1. MARV Quality Control-High

The first-generation MARV QC-High (Lot Number BMIMARV102) was created by pooling sera from three convalescent MARV infection survivors received from USAMRIID and diluting with naïve human serum (Appendix A). Based on the results from 308 independent evaluations in the MARV ELISA, BMIMARV102 was assigned an anti-MARV GP IgG concentration of 508.63 ELISA units/mL and an acceptance range (mean ± two standard deviations) of 254.31 to 762.94 ELISA units/mL. The second-generation MARV QC-High (Lot Number BMIMARV105) was also a pool of sera from two convalescent MARV infection survivors received from USAMRIID that was diluted with naïve human serum (Appendix A). Based on the results from 119 independent evaluations in the MARV ELISA, BMIMARV105 was assigned an anti-MARV GP IgG concentration of 464.13 ELISA units/mL and an acceptance range of 280.46 to 647.80 ELISA units/mL. The third-generation MARV QC-High (Lot Number BMIMARV108) was created by diluting purified human IgG polyclonal anti-MARV antibodies derived from a Tc bovine vaccinated with MARV VLP (Musoke strain) antigens (IBT BioServices Cat# 0566-001) (SAB-170M Anti-Marburg Antibody; Lot #RM1801292MV) with naïve human serum. Based on the results from 126 independent evaluations in the MARV ELISA, BMIMARV108 was assigned an anti-MARV GP IgG concentration of 427.58 ELISA units/mL and an acceptance range of 218.88 to 636.28 ELISA units/mL.

#### 2.6.2. MARV Quality Control-Low

The first-generation MARV QC-Low (Lot Number BMIMARV103) was created by pooling sera from two convalescent MARV infection survivors received from USAMRIID and diluting with naïve human serum (Appendix A). Based on the results from 306 independent evaluations in the MARV ELISA, BMIMARV103 was assigned an anti-MARV GP IgG concentration of 121.48 ELISA units/mL and an acceptance range of 60.74 to 182.22 ELISA units/mL. The second-generation MARV QC-Low (Lot Number BMIMARV106) was also a pool of sera from two convalescent MARV infection survivors received from USAMRIID that was diluted with naïve human serum (Appendix A). Based on results from 114 independent evaluations in the MARV ELISA, BMIMARV106 was assigned an anti-MARV GP IgG concentration of 138.21 ELISA units/mL and an acceptance range of 77.08 to 199.34 ELISA units/mL. The third-generation MARV QC-Low (Lot Number BMIMARV109) was created by diluting purified human IgG polyclonal anti-MARV antibodies derived from a Tc bovine vaccinated with MARV VLP (Musoke strain) antigens (IBT BioServices Cat# 0566-001) (SAB-170M Anti-Marburg Antibody; Lot #RM1801292MV) with naïve human serum. Based on the results from 126 independent evaluations in the MARV ELISA, BMIMARV109 was assigned an anti-MARV GP IgG concentration of 156.09 ELISA units/mL and an acceptance range of 83.75 to 228.44 ELISA units/mL.

#### 2.6.3. SUDV Quality Control-High

The first-generation SUDV QC-High (Lot Number BMISUDV102) was created by pooling sera from two convalescent SUDV infection survivors received from USAMRIID (not further diluted with naïve human serum, Appendix A). Based on the results from 109 independent evaluations in the SUDV ELISA, BMISUDV102 was assigned an anti-SUDV GP IgG concentration of 212.57 ELISA units/mL and an acceptance range of 99.49 to 325.65 ELISA units/mL. The second-generation SUDV QC-High (Lot Number BMISUDV105) was created by diluting purified human IgG polyclonal anti-SUDV antibodies derived from a Tc bovine vaccinated with SUDV VLP (Gulu strain) antigens (IBT BioServices Cat# 0567-001) (SAB-170S Anti-Sudan Antibody; Lot # RM1801341SV) with naïve human serum. Based on the results from 49 independent evaluations in the SUDV ELISA, BMISUDV105 was assigned an anti-SUDV GP IgG concentration of 294.84 ELISA units/mL and an acceptance range of 203.62 to 386.06 ELISA units/mL.

#### 2.6.4. SUDV Quality Control-Low

The first-generation SUDV QC-Low (Lot Number BMISUDV103) was created by pooling sera from two convalescent SUDV infection survivors received from USAMRIID and diluting with naïve human serum (Appendix A). Based on the results from 109 independent evaluations in the SUDV ELISA, BMISUDV103 was assigned an anti-SUDV GP IgG concentration of 69.34 ELISA units/mL and an acceptance range of 36.66 to 102.01 ELISA units/mL. The second-generation SUDV QC-Low (Lot Number BMISUDV106) was created by diluting purified human IgG polyclonal anti-SUDV antibodies derived from a Tc bovine vaccinated with SUDV VLP (Gulu strain) antigens (IBT BioServices Cat# 0567-001) (SAB-170S Anti-Sudan Antibody; Lot # RM1801341SV) with naïve human serum. Based on the results from 50 independent evaluations in the SUDV ELISA, BMISUDV106 was assigned an anti-SUDV GP IgG concentration of 128.13 ELISA units/mL and an acceptance range of 89.55 to 166.71 ELISA units/mL.

### 2.7. Negative Control Serum

The Negative Control serum used in the ELISA is species-matched.

#### 2.7.1. Human Negative Control Serum

NC Lot BMI530 was generated by pooling equivalent volumes of 23 individual human serum samples from Innovative Research (Novi, MI, USA) (0.00 ELISA units/mL). Other pooled human NC lots generated were BMI529 (pool of seven individual human serum samples, Innovative Research) and BMI547 (pool of nineteen individual human serum samples, Innovative Research). All NCs are used at a 1:50 single-point dilution.

#### 2.7.2. NHP Negative Control Serum

NC Lot BMI300 was generated by pooling sera from 255 individual naïve NHPs that tested negative in the MARV and SUDV ELISAs (0.00 ELISA units/mL). Other pooled NHP NC lots generated were Lot BMI298 (pool of 269 individual NHP serum samples) and BMI304 (pool of seventeen individual NHP serum samples). Both NCs are used at a 1:50 single-point dilution.

### 2.8. ELISA Optimization

#### 2.8.1. Optimization of rGP Coating Concentration

The rGP coating concentration was optimized by evaluating rGP concentrations of 0.1 to 1.0 µg/mL diluted in 1X phosphate buffered saline (PBS) with serum positive for anti-MARV or anti-SUDV IgG and a naïve human serum sample. The various rGP concentrations were coated onto a microtiter plate, one concentration per row, and a two-fold dilution series of the positive serum at a starting dilution of 1:50 and a single point of the naïve serum at a 1:50 dilution were evaluated. The OD values from the positive sample wells were used to generate 4PL curves for each rGP concentration with a conjugate dilution of 1:10,000, which is the optimum dilution for the EBOV ELISA.

#### 2.8.2. Optimization of Conjugate Dilution

The HRP-conjugated goat anti-human IgG antibody conjugate dilution was optimized using a checkerboard titration strategy in which conjugate dilutions from 1:10,000 to 1:22,000 in 1:2000 increments for MARV and from 1:6000 to 1:12,000 in 1:1000 increments for SUDV were evaluated in separate rows of a microtiter plate, and dilutions of the MARV and SUDV positive human serum and human naïve sera were evaluated in the columns of the plate. The optimal rGP coating concentration of 0.5 µg/mL (50 ng/well) was used. OD values were used to calculate binding ratios by dividing the OD values from the RS serum by the corresponding OD values for the naïve serum.

### 2.9. ELISA Qualification

#### 2.9.1. Limit of Background Cutoff Assessment

A total of 150 naïve human serum samples (50 from Innovative Research, Inc. (Novi, MI, USA) and 100 from Focus Diagnostics, Inc. (Cypress, CA, USA)) and 150 individual naïve NHP serum samples (42 from Merck and 108 in-house samples) were evaluated in duplicate at a single 1:50 dilution in both the MARV and SUDV ELISAs. The arithmetic average OD value for each sample was calculated, and a mixed-effects ANOVA model was fitted to the untransformed average values. The model was used to develop 95% prediction intervals.

#### 2.9.2. Qualification Test Sample Generation

Prior to the performance of qualification testing, the following Qualification Test Samples (QTSs) were prepared, separated into two approximately equal-volume aliquots, and stored at ≤−70 °C until use. All parent samples were coded, and no personally-identifiable information was provided. The authors did not interact with the subjects or have access to identifiable data.

**Dilutional Linearity QTSs.** A panel of five human and five NHP serum samples positive for anti-MARV or anti-SUDV GP IgG (parent samples) (Appendix A) were diluted in NC serum Lot number BMI547 (human) or BMI304 (NHP) to various levels to generate a seven-step dilution series for each parent sample (ten QTSs per parent sample, with three QTSs used to evaluate starting plate dilution). Thus, there was a total of 100 dilutional linearity QTSs) for each virus strain. An example dilutional linearity series for one parent sample is presented in Table 1. The actual dilution series generated for all parent samples (human and NHP) are presented in Appendix A.

**Specificity QTSs**. A panel of 10 individual positive samples (five human and five NHP) were used to generate a panel of forty specificity QTSs. Each individual sample was pre-adsorbed with rGP antigen (at 25 µg/mL and 10 µg/mL), CMV antigen (at 25 µg/mL), or ELISA Diluent (mock), i.e., four QTSs per individual sample. Each QTS was subjected to its assigned condition and incubated at 37 °C for 60 min ± 5 min. Following this incubation, the QTS was tested in the ELISA following the normal procedures.

#### 2.9.3. Qualification Statistical Analysis

There were four separate statistical analyses performed, one for each of the species/virus combinations. ELISA concentrations were calculated for each test sample from a minimum of 8 independent evaluations (2 independent analyses × 4 operators); however, some QTSs contributing to repeatability and within operator/between plate precision were evaluated a minimum of 16 independent times. Results were log-transformed, and a preliminary outlier analysis was performed. Once any identified outliers were removed, the following statistical analyses were performed:

**Limit of Detection:** A logistic regression analysis was performed to predict the probability that the ELISA concentration could be determined as a function of its observed geometric mean concentration. The LOD was then estimated from the model as the lowest observed mean concentration with at least 95% probability of determination.

**Limits of Quantitation:** The ULOQ and LLOQ represent the concentrations that bound the range of values for which dilutional linearity and precision are demonstrated. The final LLOQ was calculated as the median of the five corresponding averages of the last dilution level with acceptable precision and dilutional linearity from each parent sample dilution series. The ULOQ was defined as the maximum mean concentration across the five parent sample dilution series for which precision and dilutional linearity were acceptable.

**Dilutional Linearity:** A random coefficient regression model was fit to the data from the Dilutional Linearity QTSs with acceptable precision (≤50 %CV) to relate the log-transformed ELISA concentration to the log-transformed final dilution across all positive test samples. Under perfect dilutional linearity, the slope of the regression line would be exactly −1.00. The slope of the regression line across all test samples, along with a 90% confidence interval for the slope, was estimated. An equivalence analysis was performed to determine whether the assay had acceptable accuracy as measured by dilution linearity. Specifically, the assay had an acceptable accuracy relative to spike level if the 90% confidence interval was entirely contained within the acceptance interval of (−1.20, −0.80). The acceptance interval was developed based on the validation of the EBOV ELISA.

**Precision:** A mixed effects ANOVA model was fitted to the log-transformed reportable values to assess the intermediate precision, repeatability, and total assay variability (sum of intermediate precision and repeatability). The model included random effects for test day, test operator, and plate nested within operator and test day. Variability for the random effects as well as intermediate precision, repeatability, and total assay variability was estimated using the model-based percent coefficient of variation (CV). The %CV for each source of variance was calculated as $100\times \sqrt{e^{\ln{\left(10\right)}^{2}\times \sigma^{2}}-1}$, where *σ*^2^ is the model-estimated variance for the specific variance source. The percent CV associated with the model residual is an estimate of replicate observations of the same sample on the same plate evaluated by the same operator on a given test day, and hence serves to estimate repeatability. The percent CV associated with the test day, operator, and plate effects served as an estimate for the intermediate precision of the assay. Total assay variability was estimated using all variance components from the model.

To evaluate precision, the ANOVA model described above was first fit to data separately by QTS. The percent CVs for each QTS were reviewed to determine the preliminary LLOQ of the assay as those QTSs with sufficiently small and similar percent CV estimates (i.e., total assay variability not exceeding 50%). The geometric mean concentration at the lowest concentration QTS with an acceptable percent CV was estimated for each parent sample. A preliminary LLOQ was defined as the median of these five concentrations. A preliminary ULOQ was defined as the maximum geometric mean concentration with a CV percent of less than 50 percent.

All data from the dilutional linearity samples with acceptable precision were combined into a single model to evaluate precision across the range of the assay. This combined model included an additional fixed effect for QTS.

**Specificity:** The specificity analysis was restricted to the QTSs that were adsorbed with mock diluent, rGP antigen (two levels), or CMV antigen (10 test samples (5 human and 5 NHP), 4 conditions each). The difference of the log-transformed mock concentration and log-transformed rGP antigen absorbed or CMV absorbed concentration were calculated for each of the five test samples for each species separately. The mean difference across the five test samples and a corresponding 90 percent confidence interval on this mean were calculated. The results were back-transformed to the observational scale to obtain a ratio of geometric means. Equivalence tests were performed to determine if the mean concentration for each of the rGP25, rGP10, and CMV conditions was equivalent to that at the baseline (Mock).

### 2.10. Pseudovirion Neutralization Assay Procedure

The PsVNA procedure is similar to the methods used by IBT Bioservices [57], with modifications primarily to include a serum neutralization reaction of the PsV. The PsVNA was initiated by seeding VERO E6 cells at 60,000 cells per well in black walled opaque 96-well plates (Corning, Cat No. 3916). The following day test serum was diluted 1:4 (human) or 1:10 (NHP) into maintenance media (Eagle’s Minimum Essential Medium, 10% heat inactivated fetal bovine serum (HI-FBS), and 1% antibiotic/antimycotic) in Row A of a 96-well plate (Figure 3). A six-fold (human) or four-fold (NHP) serial dilution was performed down the plate.

The MARV PsV was diluted (in accordance with titration experiments performed for each PsV lot) into maintenance media with pooled human complement at 10% of the total volume (Cedarlane, Cat No. CL6600-50). The SUDV PsV dilution was similar but did not require complement. The diluted PsV was then added to all wells except the Cell Culture Control wells (wells A12, B12, C12, and D12 in Figure 3), which contained only maintenance media. The addition of PsV doubled the existing volume in each well and thus the dilution of the sera, creating a 1:8 (human) or 1:20 (NHP) starting dilution in the assay. The neutralization reaction mixture was then incubated for 60–75 min at RT. Medium from the plates containing VERO E6 cells was removed and the 100 µL of the neutralization reaction mixture was transferred onto the cells. The cells with the neutralization reaction mixture were incubated at 37 ± 2 °C with 5 ± 2% CO_2_ for 60–75 min to allow PsV entry into cells. One hundred (100) µL of pre-warmed maintenance media was added to each well of the cell-coated plates and incubated at 37 ± 2 °C with 5 ± 2% CO_2_ for 16–26 h. The neutralization reaction mixture was discarded, and cells were lysed with 1X passive lysis buffer (Promega, Cat No. E1941) and shaking at approximately 200 revolutions per minute (RPM) at RT for 30–35 min. RLU values were determined using a BioTek Synergy HTx microplate reader. The reagent injection module was set to inject 60 µL of the Working Luciferase Reagent (Promega, Cat. Nos. E152B and E151C), pause for two seconds, and then integrate luminescence for ten seconds at a gain setting of 240 for each well of the 96-well plate. A graphic of the general PsVNA method is presented in Figure 4.

The reportable values for the PsVNA (PsVNT_50_, and PsVNT_80_) were calculated using the PsVNA module within BioAssay (a proprietary Battelle software). This module used the RLU values obtained from the luminescence readings in each well and performed the following:1.The percent neutralization was determined by using the average RLU values for Cell Culture Control readings (background) as the 100% neutralization reference, and the average PsV Control as the 0% neutralization reference. The average Cell Culture Control (100% neutralization) and PsV Control (0% neutralization) RLU values were specific to the test samples on the same plate. Therefore, these averages were calculated for each plate.2.The final dilutions of serum in the PsVNA (1:8, 1:48, 1:288, 1:1728, 1:10,368, 1:62,208, 1:373,248, 1:2,239,488 for human MARV and SUDV PsVNAs or 1:20, 1:80, 1:320, 1:1280, 1:5120, 1:20,480, 1:81,920, and 1:327,680 for NHP MARV and SUDV PsVNAs) and the corresponding RLU values for each test sample were transformed to a log scale.The data were then normalized to the 0% and 100% neutralization references (average RLU values for the PsV Control and Cell Culture Control, respectively) using the following equation:3.Norm Neut_well_ = 1 − ((RLU_well_ − Mean RLU_Cell Culture Control_)/(Mean RLU_PsV Control_ − Mean RLU_Cell Culture Control_))4.A percent neutralization curve was generated using a 4PL model.5.Values for 50% and 80% neutralization were calculated by interpolating unknowns from the 4PL curve.6.The log transformed titers were then converted to a standard titer number by transforming it to log 10^(Interpolated Value) (X = 10^X).

The LOD for the PsVNA was not determined empirically but was assigned a value of 20 (human MARV and SUDV PsVNAs) or 8 (NHP MARV and SUDV PsVNAs), as it is the dilution factor for the lowest dilution tested. Samples which are <LOD were assigned a value of ½ the LOD, which is 10. The lower limit of quantitation (LLOQ) has not been determined for the PsVNA. However, from experience it has been observed that samples approximately 1- to 3-fold above the LOD may be more variable and have replicates that are <LOD. Thus, neutralization titers near the LOD (within 3-fold of the LOD) are reported and flagged as nLOD to indicate that the neutralization titer reported is near the LOD and may be below a hypothetical LLOQ.

### 2.11. Pseudovirion

The pseudovirion (PsV) is a replication deficient vesicular stomatitis virus (VSV) containing a firefly luciferase reporter gene. This PsV, which lacks the gene encoding the VSV envelope glycoprotein, G (VSV-G), was provided the MARV Angola envelope glycoprotein (MARV PsV) or the SUDV Boniface envelope glycoprotein (SUDV PsV) in trans by transfecting cells with a DNA plasmid encoding the relevant glycoprotein. MARV and SUDV PsVs were produced at IBT BioServices (Rockville, MD, USA), Catalog numbers 1004-001 and 1002-001, respectively. Lot numbers used in the development and characterization of these assays were 1809001 (human and NHP MARV PsVNAs) and 1910002 (human and NHP SUDV PsVNAs). Subsequent lots of MARV and SUDV PsVs have been characterized via titration and bridging experiments.

### 2.12. PsVNA Postitive Controls

The positive controls used in the four PsVNAs are species- and virus-specific. These positive controls are either antibodies against the virus of interest diluted into pooled human sera or pooled sera from vaccinated cynomolgus macaques. Acceptance ranges for the controls (PsVNT50 and PsVNT80) were determined during assay characterization. The controls are included on each plate to confirm the assay is performing as expected and is used to pass or fail individual plates. The human MARV PsVNA positive control (MARV-Pos-110719) was generated by diluting a rabbit anti-MARV GP antibody into a pool of naïve human sera. The NHP MARV PsVNA (022620-NHP) positive control was generated by pooling sera from NHPs vaccinated with Ad26.Filo (tri-valent filovirus vaccine (MARV, SUDV, and EBOV) based on a replication incompetent adenovirus vector), MVA-BN™-Filo (tri-valent filovirus vaccine using a Modified Vaccinia Ankara vector), and Tri-val N4CT1GP(a1) (tri-valent filovirus vaccine using an attenuated recombinant vesicular stomatitis virus vector), all of which were engineered to express MARV, SUDV, and EBOV GPs. The human SUDV PsVNA positive control (BMISUDV104) was generated by diluting purified antibodies from a SUDV-vaccinated transchromosomal bovine into a pool of naïve human sera. The NHP SUDV PsVNA positive control (NHP-SUDV-PC) was generated by pooling sera from NHPs vaccinated with Ad26.Filo, MVA-BN™-Filo, and Tri-val N4CT1GP(a1), all of which express SUDV GP, as mentioned above.

### 2.13. PsVNA Negative Controls

The negative controls used in the PsVNAs are species-specific and are pooled naïve sera from humans or NHPs. These controls were confirmed, in their respective PsVNAs, to produce a PsVNT50 value below the limit of detection (<LOD). This control is used on each plate to confirm the assay is performing as expected and is used to pass or fail individual plates. The negative control for the human MARV and human SUDV PsVNAs (BMI547) is a pool of naïve human serum samples purchased from Innovative Research, Inc. (Novi, MI, USA). The negative controls for the NHP MARV and NHP SUDV PsVNAs (BMI289 and BMI300, respectively) are pools of naïve sera from NHPs.

### 2.14. PsVNA Characterization

Characterization experiments were performed following development for each PsVNA to determine acceptance criteria and the variability of the assay. These experiments were designed to evaluate a single PsV lot or PsVNA when testing, at minimum, the negative control, positive control, PsV only control, and cell culture only control to evaluate assay performance. Surrogate, species-specific test samples were also included in the characterization efforts described above to determine the overall variability of test samples in each PsVNA. These experiments were performed over four testing days to acquire data from the controls, from which acceptance criteria were established. Three test operators were used per day with two operators running a single plate and the third operator running three plates. The operator running three plates was different on each day. Acceptance criteria for the controls were established after the statistical analysis of the data from the characterization experiments.

The reportable values used in the statistical analysis were the PsVNT_50_ and PsVNT_80_. Results from data producing an aberrant curve (data from which the 4PL curve is atypical and does not correspond to data points along the upper asymptote of the neutralization curve) and outliers via Grubbs’ test were excluded from the analysis. RLU values were analyzed for the cell culture and PsV controls.

The positive and negative controls were run in their usual plate positions (Column 10 for the negative control and Column 11 for the positive control) as well as in at least one additional test sample column (Columns 1–9). T-tests were performed to assess whether there were significant differences in either PsVNT_50_ or PsVNT_80_ values between the control data obtained from the regular control columns (Columns 10 and 11) and those obtained from the extra test sample columns. Control performance was similar regardless of plate position and, as such, the data from both sets of columns were used to determine acceptance criteria. The mean and standard deviation were calculated for each assay control. The acceptance criterion for each control and endpoint was calculated as the mean +/− two standard deviations.

The mean and standard deviation were calculated for each surrogate test sample. A mixed effects ANOVA model was fit to the log-transformed titers to assess intermediate precision, repeatability, and total assay variability (sum of intermediate precision and repeatability) overall and by test sample. The model included random effects for test day, test operator, and plate. A fixed effect for the test sample was also included. Random effects as well as intermediate precision, repeatability, and total assay variability were estimated using model-based percent coefficient of variation (CV). The %CV for each source of variance was calculated using the equation $\%\;CV=100\;x\;\sqrt{e^{\ln{\left(10\right)}^{2}\;x\;\sigma^{2}}-1}$, where *σ*^2^ is the model-estimated variance for the specific variance source.

The %CV associated with the model residual is an estimate of replicate observations of the same sample on the same plate evaluated by the same operator on a given day, and hence served to estimate the assay repeatability. The %CV associated with the test day, plate, and operator effects served as an estimate for the intermediate precision of the assay. Total assay variability was estimated using all variance components from the model (both inter- and intra-run variability). Variability (%CV) was also estimated separately by test sample. Model diagnostics were evaluated to verify assumptions regarding normality and constant variance.

## 3. Results

### 3.1. Anti-MARV and Anti-SUDV GP IgG ELISA Development

The anti-MARV and anti-SUDV anti-GP IgG ELISAs were developed following the same method as described for the validated anti-EBOV GP IgG ELISA [56] with changes in materials specific for MARV and SUDV (e.g., coating antigen, reference standard (RS), quality controls (QCs), starting dilutions). All other aspects (incubation times and temperatures, secondary antibody/conjugate, substrate, stop solution, buffers) were the same. However, these shared reagents and parameters were confirmed to be acceptable and optimized if necessary. Due to the conjugate being fully cross-reactive with sera from both humans and NHPs, these assays use RS and QCs from humans for analysis of both human and NHP samples, except the negative control (NC), which is species-matched.

#### 3.1.1. Optimization of rGP Coating Concentration

First, the recombinant GP (rGP) coating concentration was optimized. The choice of rGP strain to use as the coating antigen is important in that it should match the strain used in the nonclinical efficacy studies [10]. The rGP strain used for the MARV ELISA was Ci67 and for the SUDV ELISA was Gulu. Unfortunately for the MARV ELISA, the MARV Angola strain is being used as the challenge agent for the nonclinical studies. However, there are efforts underway to qualify the rGP MARV Angola strain for use in the ELISA. This involves the requalification of the RS and reestablishment of the acceptance ranges for the QCs.

The optimal coating concentration for rGP (MARV Ci67 and SUDV Gulu) was assessed over various coating concentrations from 0.1 to 1.0 µg/mL using a positive convalescent human sample in both the MARV and SUDV ELISAs. The conjugate dilution used for these experiments was 1:10,000, which was the optimum dilution used in the EBOV ELISA. Data for the SUDV ELISA are shown in Appendix A and the data for MARV ELISA are shown in Appendix A. As expected, the OD values were higher for the immune serum (Columns 1 through 10) than for the naïve serum (Column 12) at all rGP concentrations tested. Furthermore, as observed in Column 11 (serum blank), the conjugate did not bind to rGP antigen in the absence of serum, suggesting that it does not bind non-specifically to rGP on the plate.

Dose response curves (4 parameter logistic model, 4PL) for the positive sample showed a dilution effect for all coating antigen concentrations tested (Figure 5 (SUDV) and Figure 6 (MARV)). An increase in OD values was observed as the concentration of the coating antigen increased. The desired maximum OD value of approximately 2.7 at the 1:100 dilution of the test sample was achieved at an antigen coating concentration of 0.5 µg/mL with the addition of 100 µL/well (50 ng per well). Thus, 0.5 µg/mL (100 µL/well; 50 ng/well) was chosen as the optimized rGP coating concentration for the human MARV and SUDV ELISAs. NHP serum was not tested, but a rGP concentration of 0.5 µg/mL was also selected for testing NHP samples.

#### 3.1.2. Optimization of Conjugate Dilution

Next, the conjugate dilution was optimized. The OD values and binding ratios are presented in the Appendix A. The binding ratios in the blank row (Row H) are approximately one, suggesting that background is about the same in both the positive plate and negative plate. Additionally, the OD values for the 1:50 dilution of the positive serum were near 3.0, while the OD value for the naïve serum was low (less than 0.1). Based on depth of curve, binding ratios, and the fact that a 1:10,000 dilution is used for the EBOV ELISA, the optimal dilution of the conjugate was selected to be 1:10,000 for both the MARV and SUDV ELISA

#### 3.1.3. Creation of Serum Controls

Panels of sera from the convalescent survivors of a previous MARV or SUDV infection were used in the creation of the first-generation critical reagents (RS and QCs). The first-generation RS for the MARV ELISA was Lot BMIMARV101 and for the SUDV ELISA the lot was BMISUDV101; each was a pool of sera from two survivors (Appendix A). The optimum RS starting dilution and dilution schemes were chosen using data generated using different starting dilutions (e.g., 1:50 and 1:100) and dilution schemes (e.g., 1:2, 1:1.9, 1:1.8, 1:1.7, 1:1.6, and 1:1.5) and fitted to a 4PL model. The optimal starting dilution for BMIMARV101 was selected as 1:100 and BMISUDV101 was selected as 1:50 with a seven-step 1:1.7 dilution scheme for both (Figure 7 (MARV) and Figure 8 (SUDV)). For each of these RSs, the resulting 4PL curve had well-defined upper and lower asymptotes and typically five dilution points in the linear range of the curve. Subsequent generations of RS were created and further described in the Materials and Method section.

The first-generation QCs (QC-High and QC-Low) for the MARV and SUDV ELISAs were created by pooling sera from convalescent survivors (Appendix A), followed by evaluation in their respective ELISAs of more than 300 independent times. BMIMARV102 (QC-High) and BMIMARV103 (QC-Low) had average concentrations of 508.63 ELISA units/mL and 121.48 ELISA units/mL, respectively. BMISUDV102 (QC-High) and BMISUDV103 (QC-Low) had average concentrations of 212.57 ELISA units/mL and 69.34 ELISA units/mL, respectively. Subsequent generations of QCs (three generations total for MARV and two generations total for SUDV) were created and further described in the Materials and Method section.

Negative Control (NC) was also generated, characterized, and qualified for use in the ELISAs. Twenty-five unique human serum samples were evaluated in the optimized ELISA and twenty-three had a mean OD value less than or equal to 0.20, which is the OD acceptance criterion for an NC. These twenty-three samples were pooled into a single volume that was designated as NC Lot number BMI530. Subsequent lots of NC (both human and NHP) were created and further described in the Materials and Method section.

### 3.2. Anti-MARV and Anti-SUDV GP IgG ELISA Qualification

Following the optimization of the ELISA method and qualification of the critical reagents, ELISA performance was characterized through qualification testing for both the anti-MARV and anti-SUDV GP IgG ELISAs. These qualifications assessed the limit of detection (LOD), the limits of quantitation, repeatability, intermediate precision, and dilutional linearity. For each qualification, 50 human and 50 NHP qualification test samples were generated from five parent samples each that were diluted in naïve human or NHP serum to various levels to generate a panel of samples spanning the dynamic range of the assay. Qualification samples were evaluated multiple times each, as described in the Qualification Statistical Analyses section.

For the qualification results, an outlier analysis identified five potential outliers for human and two potential outliers for NHP for the anti-MARV GP IgG ELISA (Appendix A). No test errors were associated with these observations; thus, they were included in the statistical analysis. An outlier analysis identified six potential outliers for human and four potential outliers for NHP for the anti-SUDV GP IgG ELISA (Appendix A). Although no test errors were associated with these observations, the two reported results of 0 ELISA units/mL for QTS 15 and 18 as well as the high observation for QTS 16 were excluded from the statistical analysis; the other observations were included in the statistical analysis. Following outlier analysis, the results were statistically analyzed as described in the Qualification Statistical Analyses section. A summary of the qualification results is provided in Table 2 (MARV) and Table 3 (SUDV).

#### 3.2.1. Limit of Background Cutoff Assessment

To generate an OD value cutoff for scoring samples as negative in the ELISA, a mixed-effects analysis of variance (ANOVA) model was fitted to the average OD values from 150 naïve human and naïve NHP serum samples tested in the ELISA, with each species evaluated separately. The 95% prediction intervals when a sample is tested at a dilution of 1:50 were determined to be a mean OD value of 0.000–0.107 for human and 0.000–0.050 for NHP in the anti-MARV GP IgG ELISA (Table 2); and a mean OD value of 0.000–0.134 for human and 0.000–0.384 for NHP in the anti-SUDV GP IgG ELISA (Table 3). The upper bounds of these intervals serve as the limit of background for each species/virus combination and are used to score samples as negative.

#### 3.2.2. Limit of Detection

The LOD is the lowest antibody concentration for which there is at least 95% probability that an estimate can be obtained. From the logistic curve comparing the probability of detection against the geometric mean observed values for all test samples, the LOD for the anti-MARV GP IgG ELISA was estimated to be 11.42 ELISA units/mL (human) and 11.67 ELISA units/mL (NHP) (Figure 9 and Table 2) and the LOD for the anti-SUDV GP IgG ELISA was estimated to be 7.15 ELISA units/mL (human) and 10.24 ELISA units/mL (NHP) (Figure 9 and Table 3).

#### 3.2.3. Limits of Quantitation

Quantitation limits for an assay establish the ranges in which dilutional linearity and precision are acceptable. The lower limit of quantitation (LLOQ) for the MARV ELISA was determined to be 76.67 ELISA units/mL (human) and 20.42 ELISA units/mL (NHP), and the upper limit of quantitation (ULOQ) was determined to be 5232.12 ELISA units/mL (human) and 6898.26 ELISA units/mL (NHP) (Table 2). The LLOQ for the SUDV ELISA was determined to be 11.56 ELISA units/mL (human) and 18.55 ELISA units/mL (NHP), and the ULOQ was determined to be 11,542.60 ELISA units/mL (human) and 17,199.09 ELISA units/mL (NHP) (Table 3). The ULOQs are conservative estimates limited by the availability of high-concentration samples at the time of testing; the true ULOQs are likely to be higher.

#### 3.2.4. Dilutional Linearity

The accuracy of an analytical method describes how close the test results are to the true value. Absolute accuracy can be measured when test samples of a known quantity can be included in the assay. No such samples were available in this analysis. In these cases, accuracy may be assessed by examining the results of dilutional linearity analysis, sometimes referred to as “relative accuracy”. Dilutional linearity refers to obtaining test results over a series of dilution levels that are directly proportional to the respective dilution levels. The accuracy of the ELISA was characterized via dilutional linearity to evaluate whether the assay can obtain results that are proportional to the concentration of the antibody in a given sample. Dilutional linearity was determined through the comparison of the resulting value for a test sample to the spike level of each sample. The CV percent was calculated for each test sample and spike level (Appendix A for MARV and Appendix A for SUDV), and only spike levels that met an arbitrary maximum desired CV percent of 50% were used to evaluate dilutional linearity.

Under perfect dilutional linearity, the slope of the regression of log-transformed concentration on log-transformed spike level should be −1. For the MARV ELISA, the overall regression line across all human parent test samples was −1.01 with a 90% confidence interval of −1.04 to −0.98 and for all NHP parent test samples was −1.02 with a 90% confidence interval of −1.04 to −1.01, showing that the assay is accurate across the range of concentrations tested (Table 2). For the SUDV ELISA, the overall regression line across all human parent test samples was −0.97 with a 90% confidence interval of −1.00 to −0.94 and for all NHP parent test samples was −0.97 with a 90% confidence interval of −1.00 to −0.94, showing that the assay is accurate across the range of concentrations tested (Table 3).

#### 3.2.5. Precision

The precision of an analytical method describes the closeness of individual measures of an analyte when the procedure is applied repeatedly to multiple aliquots of a single homogeneous volume of a test sample. To evaluate the precision of the human MARV and SUDV ELISAs, two parameters (intermediate precision and repeatability) were determined using results from the 100 qualification test samples (Table 2 and Table 3). Intermediate precision incorporates variation due to operator-to-operator, day-to-day, and plate-to-plate. Repeatability incorporates variation within a plate. The residual variance from the ANOVA model, which includes repeatability as well as any other unexplained variability not accounted for in intermediate precision, was used as an estimate for repeatability. Total assay variability was also calculated. For the MARV ELISA, the intermediate precision was calculated to be 25.0% for human samples and 21.3% for NHP samples. The %CV for repeatability was calculated to be 16.9% for human samples and 19.1% for NHP samples. The %CV for total assay variability was calculated to be 30.4% for human samples and 28.9% for NHP samples. For the SUDV assay, the intermediate precision was calculated to be 15.5% for human samples and 21.6% for NHP samples. The %CV for repeatability was calculated to be 15.0% for human samples and 14.0% for NHP samples. The %CV for total assay variability was calculated to be 21.7% for human samples and 26.0% for NHP samples.

#### 3.2.6. Specificity

The goal of testing specificity is to assess the ability of the ELISA to differentiate the analyte (anti-GP IgG) from nonspecific analytes that may also be present in the sample matrix. The specificity of each ELISA was assessed by comparing the anti-GP IgG concentrations for five positive human serum samples and five positive NHP serum samples that were adsorbed with a homologous (MARV or SUDV rGP) or heterologous (CMV) antigen prior to analysis. The CMV antigen was selected because the CMV antigen is completely unrelated to filovirus antigens, CMV is highly prevalent in both African and North American populations [58,59], and the CMV recombinant protein was generated in a manner consistent with the MARV and SUDV rGP (expression in HEK293 cells). For a given serum sample, a decrease in the detectable anti-MARV or SUDV GP IgG levels when adsorbed with the rGP but no decrease when adsorbed with the CMV antigen indicated that the antibodies detected in the non-adsorbed sample were MARV or SUDV rGP-specific.

For the MARV ELISA, no change was observed in the test samples adsorbed with the CMV antigen, with observed mean ratios of anti-MARV GP IgG in mock-adsorbed samples compared to samples adsorbed against the CMV antigen determined to be 1.13 (90% upper bound: 1.27) and 1.05 (90% upper bound: 1.10) for human and NHP samples, respectively. In contrast, the mean ratios for mock-adsorbed samples compared to adsorption with 10 µg/mL rGP was 76.58 (90% lower bound: 33.04) and 16.67 (90% lower bound: 5.33) for human and NHP samples, respectively. The mean ratios for mock-adsorbed samples compared to adsorption with 25 µg/mL rGP was 366.09 (90% lower bound: 133.77) and 174.57 (90% lower bound: 0) for human and NHP samples, respectively (Table 2, Appendix A). The ratio for the 25 µg/mL rGP was higher than the ratio for the 10 µg/mL rGP, supporting the conclusion that the ELISA is specific for anti-GP IgG.

For the SUDV ELISA, no change was observed in the test samples adsorbed with the CMV antigen, with observed mean ratios of anti-SUDV GP IgG in mock-adsorbed samples compared to samples adsorbed against the CMV antigen determined to be 1.13 (90% upper bound: 1.31) and 1.09 (90% upper bound: 1.21) for human and NHP samples, respectively. In contrast, the mean ratios for mock-adsorbed samples compared to adsorption with 10 µg/mL rGP was 3.02 (90% lower bound: 1.81) and 4.14 (90% lower bound: 1.42) for human and NHP samples, respectively. The mean ratios for mock-adsorbed samples compared to adsorption with 25 µg/mL rGP was 3.96 (90% lower bound: 1.88) and 8.61 (90% lower bound: 0) for human and NHP samples, respectively (Table 3, Appendix A). The ratio for the 25 µg/mL rGP was higher than the ratio for the 10 µg/mL rGP, supporting the conclusion that the ELISA is specific for anti-GP IgG.

### 3.3. Pseudovirion Neutralization Assay Development

The initial PsVNA procedure was based upon methods used by IBT Bioservices [57]. These methods were modified to include a neutralization reaction component to allow neutralizing antibodies in sera to bind the pseudovirion (PsV). Additional modifications to the plate layout, cell lysis procedure, and luminometer readings were incorporated. Early development experiments utilizing filovirus PsVs and serum from several species were conducted to establish the basic procedure for the PsVNA. Subsequent experiments were conducted to determine aspects of each assay that may be modified to accommodate different serum species and PsVs, such as the starting dilution, the serial dilution, the use of complement, and the amount of PsV to be used on a per plate basis.

The first PsVNA development experiments used positive immune serum and naïve serum to show that the assay was able to measure neutralization of the pseudovirion (PsV) in positive serum and not detect neutralization in naïve serum. Once the basic performance of the PsVNA procedure was verified using positive and naïve serum samples, additional aspects of the assay were evaluated, including VERO E6 cell seeding densities, luciferase reagent volumes, and the impact of mild shaking during the neutralization reaction. These experiments established a seeding density of 60,000 cells per well one day prior to conducting the PsVNA, a luciferase reagent volume of 60 µL per well, and no shaking during the neutralization reaction. This established the general assay procedure used for all PsVNAs, regardless of the PsV or serum species used (Figure 4).

Next, aspects of the PsVNA specific to the PsV (MARV and SUDV-specific) and serum species (Human and NHP) used were evaluated. The volume of PsV needed for each assay was evaluated for each PsV lot to determine the optimum PsV concentration. IBT Bioservices suggests that one vial of the PsV should be sufficient for one 96-well plate. Different volumes are provided for each PsV lot, so MARV and SUDV PsVs were tested at quantities ranging from one-ninth of a vial to one and one-third of a vial per plate, targeting RLU values of 2.00 × 10^5^–2.00 × 10^6^ in PsV only control wells. The optimal amount of MARV PsV Lot 1809001 was determined to be one-third of a plate per vial and optimal amount of SUDV PsV Lot 1910002 was one and one-third of a vial per plate.

The use of complement was examined in both the MARV and SUDV PsVNAs. Complement increased the sensitivity of both PsVNAs; however, only the MARV PsVNAs used complement since it resulted in background neutralization titers in naïve samples when used in the SUDV PsVNA. Test serum starting dilutions of 1:8, 1:20, 1:30, and 1:50 as well as serial dilutions of 1:4 and 1:6 were also evaluated. False positives were more often observed in naïve NHP samples than in naïve human samples. Thus, a 1:8 starting dilution was used for human samples because few naïve serum samples produced a 50% neutralization titer at this starting dilution. However, a 1:20 starting dilution was used for NHP samples to minimize false positive neutralization results. A 1:4 serial dilution was used for NHP samples in an effort to maintain more dilutions in the linear range of the assay, since a 1:6 serial dilution with a 1:20 starting dilution would result in unnecessarily high dilutions at the end of the dilution scheme.

Once the procedure for each PsVNA was established, species-specific positive and negative controls were established. Pools of naïve human and NHP sera were evaluated for each assay to identify an appropriate negative control that would not produce a detectable neutralization titer. To establish positive controls, neutralizing antibodies diluted into human serum and immunized NHP serum pools were evaluated in the assay.

### 3.4. Pseudovirion Neutralization Assay Characterization

Four separate sets of PsVNA characterization experiments were performed, one for each assay (human MARV and SUDV, and NHP MARV and SUDV PsVNAs). Each set of experiments was performed by three operators over four testing days. After the conclusion of the characterization experiments, the data from all four testing days were subjected to statistical analysis. These analyses were used to evaluate assay variability and determine appropriate acceptance criteria for controls and test samples.

The variability of each PsVNA is summarized in the top section of Appendix A, which show the %CV for each variance component in the model along with the intermediate precision, repeatability, and total assay variability estimates for the PsVNT_50_ and PsVNT_80_ endpoints. These variability estimates were determined after all control and surrogate test samples used for a given assay were analyzed together. Surrogate test samples are species-specific samples (i.e., serum from humans or NHPs) expected to produce a neutralization titer and represent test samples likely to be evaluated in the PsVNA. The total assay variability (intermediate precision and repeatability) ranged from 29.2% (human MARV PsVNA) to 52.0% (human SUDV PsVNA) for the PsVNT_50_ and 23.3% (NHP SUDV PsVNA) to 46.3% (human SUDV PsVNA) for the PsVNT_80_. The bottom section of Appendix A displays the %CV ranges observed for total assay variability when evaluated separately for each surrogate test sample and positive control. Variability among the individual samples and positive controls used within each of the four PsVNAs ranged from 15.4% to 59.1% for PsVNT_50_ values and from 13.8% to 64.2% for the PsVNT_80_.

Acceptance criteria for the negative, positive, PsV, and cell culture controls were established after statistical analysis of the PsVNA characterization data (Appendix A). None of the negative controls used in the four PsVNAs had a 50% or 80% neutralization titer, as expected. Representative neuralization data for each of the negative controls are shown in Figure 10. For the positive control data, the mean +/− 2 standard deviation was used to set the acceptance criteria ranges for the PsVNT_50_ and PsVNT_80_ for each PsVNA. Representative neutralization curves for each of the positive controls are shown in Figure 11. Acceptance criteria ranges provided for Relative Light Unit (RLU) values pertaining to the cell culture and PsV controls were expanded beyond the mean +/− 2 standard deviations. This was to account for variation typically observed among cell passages while maintaining a minimum RLU value for the PsV control.

Acceptance criteria pertaining to the 50% and 80% neutralization titers of test sample replicates are summarized in Appendix A. These criteria were established after considering the total assay variability and the variability of the individual test samples using the characterization data (Appendix A).

Overall, each virus and species-specific PsVNA characterization was successful in that the performance of the controls allowed for acceptance criteria ranges to be established. When performing a PsVNA to evaluate test samples, each plate must meet all acceptance criteria outlined in Appendix A, while each set of test sample replicates must meet the criteria outlined in Appendix A. These acceptance criteria establish a basic level of control for each PsVNA when evaluating serum samples.

## 4. Discussion

The progression of candidate MARV and SUDV vaccines through human clinical trials and to potential licensure via the FDA Animal Rule will be dependent on the bridging of immune correlates of protection from NHP studies with equivalent human clinical trial endpoints. As observed during the 2014–2016 Ebola disease outbreak in Sierra Leone, high levels of anti-GP IgG correlated with survival [27] and candidate vaccines for MARV and SUDV employ expression of GP as the antigen [9,11]. Thus, anti-GP IgG antibodies are the likely front runners for a correlate of protection. Successful bridging studies will be dependent upon the development and validation of species-neutral methods, such as ELISAs and neutralization assays, for measuring these potential correlates in both humans and NHPs.

To meet this demand, we have described here the development and qualification of the anti-MARV and anti-SUDV GP IgG ELISAs and shown that they are suitable for their intended purpose to measure anti-MARV and anti-SUDV GP IgG levels in human and NHP serum. Both ELISAs are sensitive (low LOD and LLOQ), precise (total assay variability ≤ 30%), dilutionally linear across their analytical range (slopes of approximately −1), and specific for anti-MARV/SUDV GP IgG in human and NHP serum. The next steps will be the full validation of both assays prior to being used for assessing the immunogenicity of candidate MARV/SUDV vaccines in human clinical trials and nonclinical NHP studies.

Additionally, we have described the development of the PsVNA and shown that it is suitable for its intended purpose to measure the levels of neutralizing antibodies in humans and NHPs for both MARV and SUDV. The assay is moderately precise, with total assay variability %CVs being between 29% and 52% for the PsVNT_50_ readout and between 23% and 46% for the PsVNT_80_ readout. It is important to note that an LOD and LLOD have not yet been empirically determined. Instead, an LOD has been assigned based on the starting dilution. All results above the LOD were used to calculate total assay variability, which may have resulted in artificially high estimates of total assay variability as the true LOD/LLOQ may be higher than estimated here. Future qualification efforts will include empirically defining the LOD and LLOQ, which will result in better estimates of precision.

In summary, the assay readouts for both the ELISA and PsVNA assays described here are strong candidates for a correlate of protection for the evaluation of candidate MARV and SUDV vaccines and should be considered when designing clinical trials and nonclinical studies for the evaluation of candidate MARV and SUDV vaccines.

## Figures and Tables

**Figure 1 vaccines-10-01211-f001:**
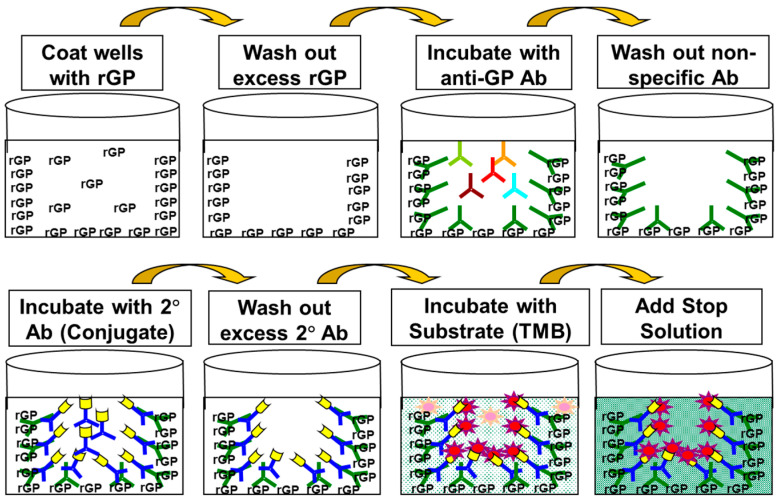
Illustration of General Anti-GP IgG ELISA Method.

**Figure 2 vaccines-10-01211-f002:**
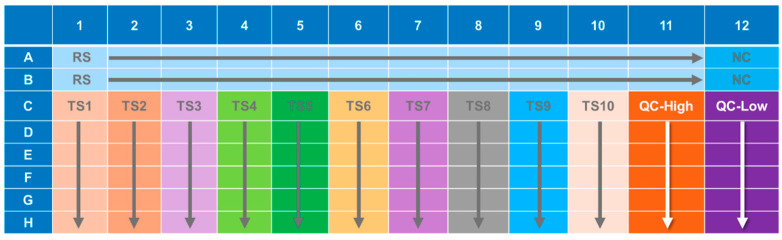
Illustration of Anti-GP IgG ELISA Plate Layout. RS = References Standard; TS# = Test Samples 1 through 10; QC-High = Quality Control—High; QC-Low = Quality Control—Low; NC = Negative Control. Arrows indicate 2-fold dilutions either across or down the plate.

**Figure 3 vaccines-10-01211-f003:**
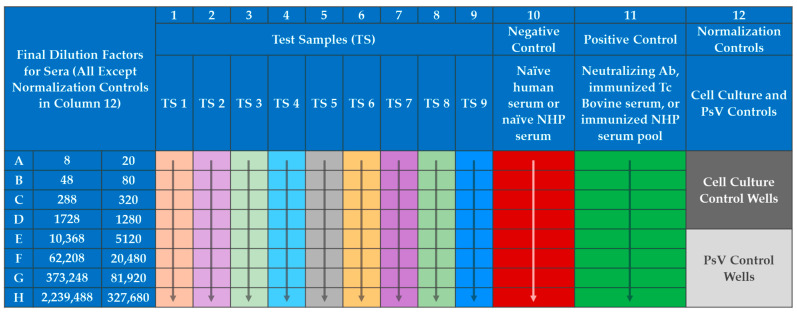
PsVNA Plate Layout.

**Figure 4 vaccines-10-01211-f004:**
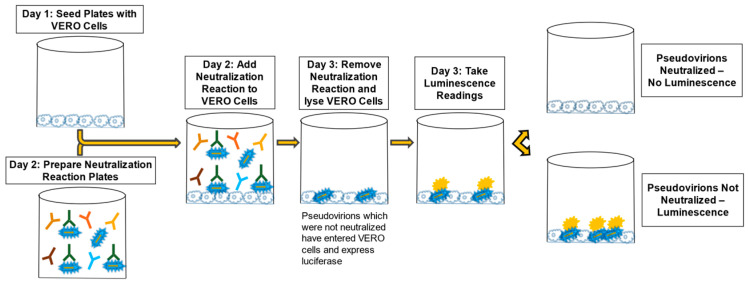
Graphic of PsVNA Method.

**Figure 5 vaccines-10-01211-f005:**
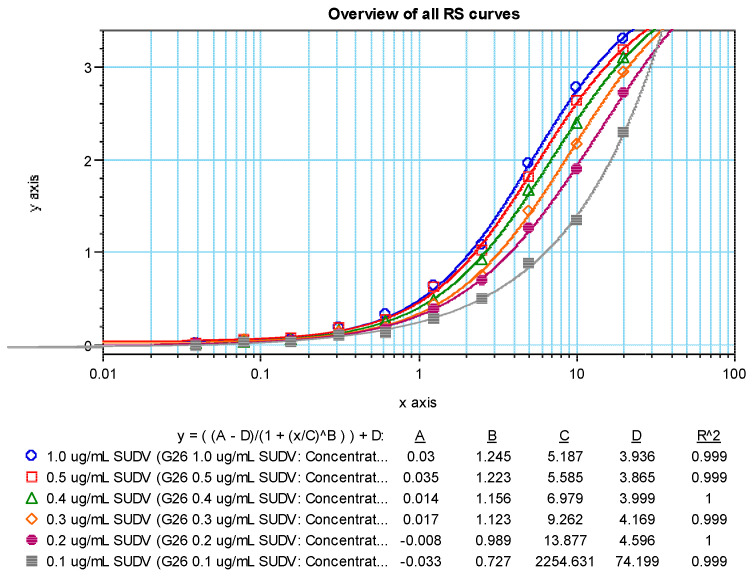
Coating Antigen Optimization 4PL Curve Fits—SUDV.

**Figure 6 vaccines-10-01211-f006:**
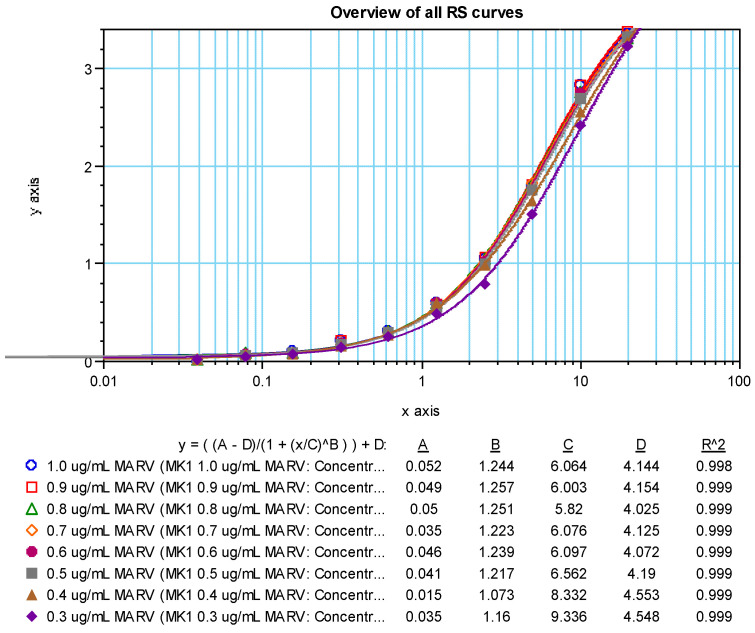
Coating Antigen Optimization 4PL Curve Fits—MARV.

**Figure 7 vaccines-10-01211-f007:**
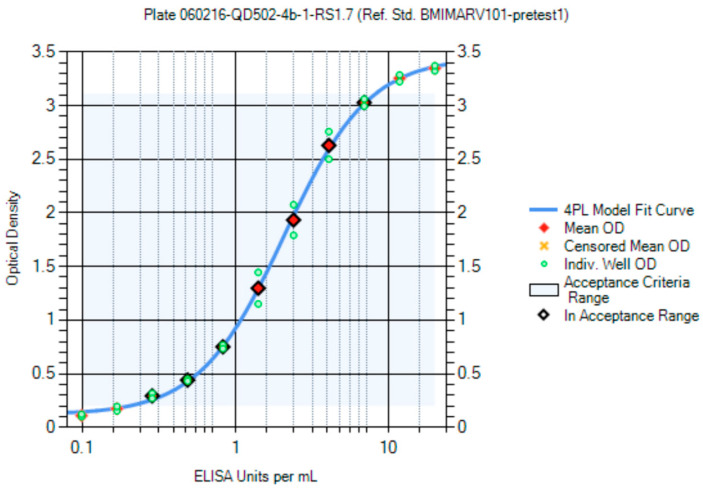
Optimization of BMIMARV101 Reference Standard Dilution Scheme. BMIMARV101 was tested at a 1:100 starting dilution (Initial 1:2 dilution with naïve serum followed by a 1:50 dilution on the plate) and 1:1.7 dilution scheme.

**Figure 8 vaccines-10-01211-f008:**
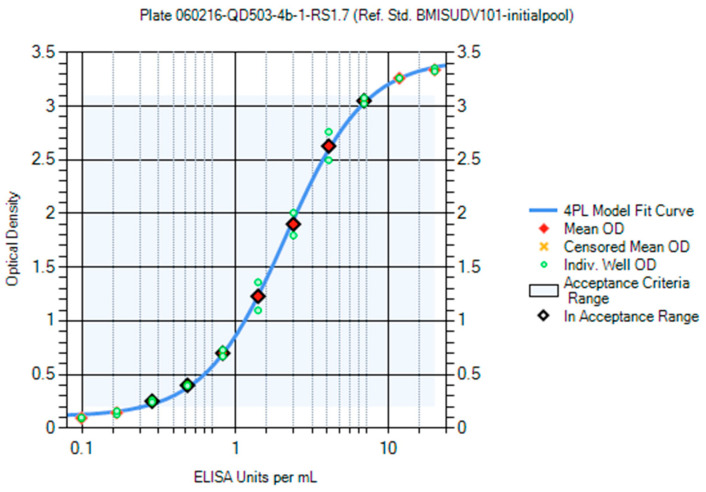
Optimization of BMISUDV101 Reference Standard Dilution Scheme. BMISUDV101 was tested at a 1:50 starting dilution and 1:1.7 dilution scheme.

**Figure 9 vaccines-10-01211-f009:**
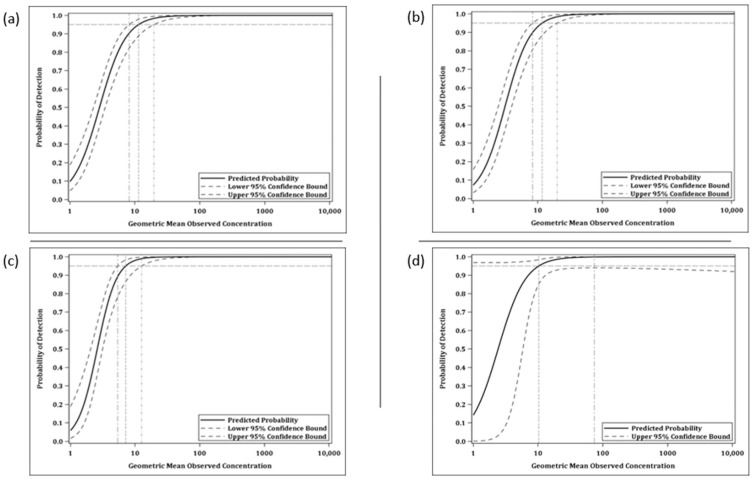
Logistic regression curve showing relationship between the probability of detecting the concentration and the mean log-transformed observed value. (**a**) MARV and human; (**b**) MARV and NHP; (**c**) SUDV and human; (**d**) SUDV and NHP.

**Figure 10 vaccines-10-01211-f010:**
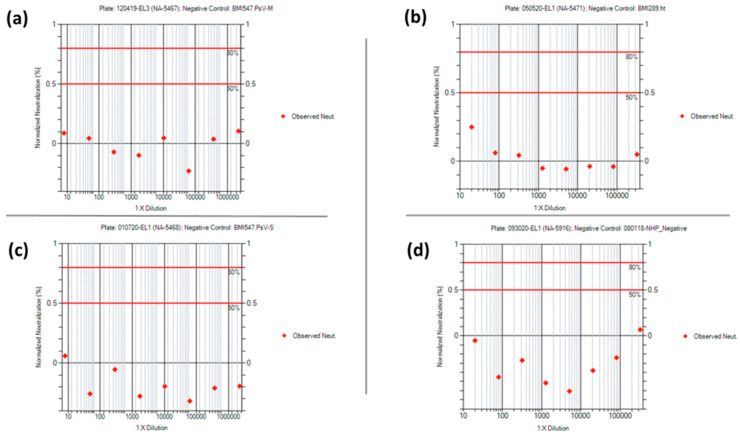
PsVNANeutralization Data from Negative Controls. (**a**) MARV and human; (**b**) MARV and NHP; (**c**) SUDV and human; (**d**) SUDV and NHP. The negative control indicated in the SUDV and NHP panel, 080118-NHP_Negative, is a subaliquot of BMI300.

**Figure 11 vaccines-10-01211-f011:**
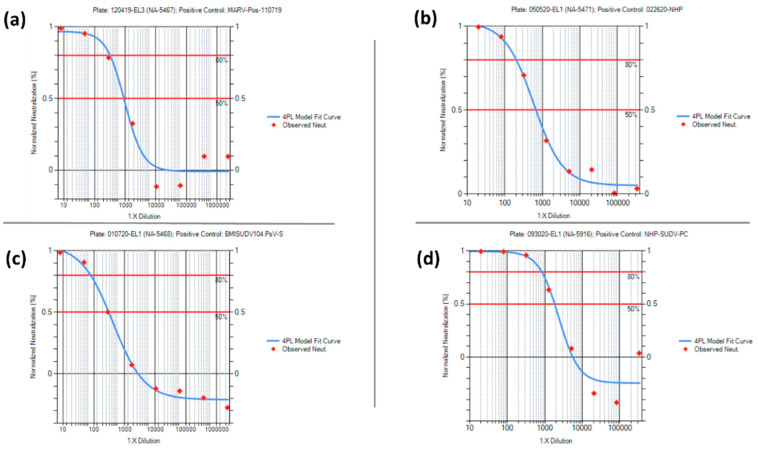
Neutralization Data from Positive Controls. (**a**) MARV and human; (**b**) MARV and NHP; (**c**) SUDV and human; (**d**) SUDV and NHP.

**Table 1 vaccines-10-01211-t001:** Example Dilution Series for One Parent Sample (FLV1001-100023 Day 71).

QTS #	Final Spike Level	Expected QTS Anti-GP IgG Concentration (ELISA Units/mL)	QTS Starting Dilution
1	1:1	1284	1:100
2	1:1	1284	1:200
3	1:1	1284	1:50
4	1:2	642	1:50
5	1:5	257	1:50
6	1:8	160	1:50
7	1:16	80	1:50
8	1:80	16	1:50
9	1:100	13	1:50
10	1:150	9	1:50

**Table 2 vaccines-10-01211-t002:** Summary of Anti-MARV GP IgG ELISA Qualification Results.

Characterization Parameter	Results	
	Human	NHP
Limit of Background (Upper Bound of 95% Prediction Interval)	Mean OD value of 0.170	Mean OD value of 0.050
Limit of Detection	11.42 ELISA Units/mL	11.67 ELISA Units/mL
Lower Limit of Quantitation	76.67 ELISA Units/mL	20.42 ELISA Units/mL
Upper Limit of Quantitation	5232.12 ELISA Units/mL	6898.26 ELISA Units/mL
Repeatability	16.9%	19.1%
Intermediate Precision	25.0%	21.3%
Total Assay Variability	30.4%	28.9%
Dilutional Linearity (Accuracy)	Slope of −1.01 (90% CI of −1.04, −0.98)	Slope of −1.02 (90% CI of −1.04, −1.01)
Specificity (Ratio and 90% Confidence Bounds)	CMV = 1.13 (1.00, 1.27) rGP 10 µg = 76.58 (33.04, 120.13) rGP 25 µg = 366.09 (133.77, 598.41)	CMV = 1.05 (1.00, 1.10) rGP 10 µg = 16.67 (5.33, 28.01) rGP 25 µg = 174.57 (<0, 402.12)

**Table 3 vaccines-10-01211-t003:** Summary of Anti-SUDV GP IgG ELISA Qualification Results.

Characterization Parameter	Results	
	Human	NHP
Limit of Background (Upper Bound of 95% Prediction Interval)	Mean OD value of 0.134	Mean OD value of 0.384
Limit of Detection	7.15 ELISA Units/mL	10.24 ELISA Units/mL
Lower Limit of Quantitation	11.56 ELISA Units/mL	18.55 ELISA Units/mL
Upper Limit of Quantitation	11,542.60 ELISA Units/mL	17,199.09 ELISA Units/mL
Repeatability	15.0%	14.0%
Intermediate Precision	15.5%	21.6%
Total Assay Variability	21.7%	26.0%
Dilutional Linearity (Accuracy)	Slope of −0.97 (90% CI of −1.00, −0.94)	Slope of −0.97 (90% CI of −1.00, −0.94)
Specificity (Ratio and 90% Confidence Bounds)	CMV = 1.13 (0.94,1.31) rGP 10 µg = 3.02 (1.81,4.23) rGP 25 µg = 3.96 (1.88,6.05)	CMV = 1.09 (0.96,1.21) rGP 10 µg = 4.14 (1.42,6.86) rGP 25 µg = 8.61 (<0,17.72)

## Data Availability

The data presented in this study are available upon reasonable request from the corresponding author.

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
