# Peer review of "Assays for the Evaluation of the Immune Response to Marburg and Ebola Sudan Vaccination—Filovirus Animal Nonclinical Group Anti-Marburg Virus Glycoprotein Immunoglobulin G Enzyme-Linked Immunosorbent Assay and a Pseudovirion Neutralization Assay"

_vaccines, 2022, doi:10.3390/vaccines10081211_

Round 1

Reviewer 1 Report

In this study, authors described development of assays for quantification of anti-Marburg virus (MARV) and anti-Sudan virus (SUDV) antibodies and for quantification of MARV and SUDV neutralizing antibodies in human and non-human primates’ samples. The assay represents the valuable tools for testing of potential vaccines against MARV and SUDV. Developed pseudovirion neutralizing assay is suitable for measuring of MARV and SUDV neutralizing antibodies in humans and non-humans primates. This test is safe and can be performed in laboratory. The chapter Materials and Methods is precisely described and the text has been supplemented by illustrations. The obtained results are well documented in the tables and figures. The manuscript is very well written, everything is described to the smallest details. The conclusions are consistent with presented results.

I have some comments:

1. It is pity that instead of number of Figure or table is the text: Error! Reference source not found. Authors should correct it.

2. The Figure 9, 10, 11. The legends in the figures are too small. The labelling of the axis is not readable.

3. Reference 12 is not correctly cited

Reviewer 2 Report

Review of Manuscript “Assays for the Evaluation of the Immune Response to Marburg and Ebola Sudan Vaccination – Filovirus Animal Nonclinical Group anti-Marburg Virus Glycoprotein Immunoglobulin G Enzyme-Linked Immunosorbent Assay and a Pseudovirion  Neutralization Assay“ by Thomas L. Rudge Jr. et al..

The development of effective Marburg virus (MARV) and Sudan virus (SUDV) vaccines to cope with sporadic filovirus outbreaks will require reliable quantitation of anti-GP IgG, as the levels of these antibodies often correlate with survival and thus may also present a valid correlate for protection from a vaccine.

In the present manuscript the authors describe in a very detailed fashion the development and qualification of the anti-MARV and anti-SUDV GP IgG ELISAs, which seem to be well suited for their intended use to reliable measure anti-MARV and anti-SUDV GP IgG levels in human and NHP serum. Both ELISAs are shown to be quite sensitive, precise and specific for anti-MARV/SUDV GP IgG in human and NHP serum.

Furthermore, the development of neutralization assays (PsVNA) to measure neutralizing antibodies against MARV and SUDV, respectively, is also described.

The development of the assays is described in a very detailed and comprehensive manner including a variety of appropriate controls. The manuscript is well-written, but may be shortened at some points for more clarity and readability (see also major points below).   

I would therefore like to recommend publication, if the major and minor points listed in detail below are addressed in a revised version of the manuscript.

Major points

1) The actual manuscript should be shortened by omitting some of the tables (such as tables 1 to 4), rather presenting these in the supplementary data

2) Tables 6 and 7 should also be relocated to the supplement, since the data is shown in a clear form in fig. 5 and 6.

3) The references to the tables and figures are not correct throughout the manuscript (statement “Error! Reference source not found“ throughout the manuscript)

Minor points:

1) In the Material and Methods part I would switch the order of sections 2.10 and 2.11 (first describe the principle of the neutralization assay and then the detailed implementation)  

2) The graphs shown in fig. 10 and 11 should be presented at a better resolution.
